# The Impact of COVID-19 Process on Sustainability in Education: Work Alienation of Physical Education and Sports Teachers

**Cenk Temel [1], Çiğdem Gökduman [1,*], Sinan Uğraş [2], Ahmet Enes Sağın [3], Mehmet Akif Yücekaya [3], Mehmet Kartal [4] and Turhan Toros [5]**

1  Department of Sport Management, Akdeniz University, Dumlupinar Boulevard, 07070 Antalya, Turkey
2  Department of Physical Education and Sports Teaching, Canakkale Onsekiz Mart University (COMU), 17100 Canakkale, Turkey
3  Ministry of Education, 06000 Ankara, Turkey
4  Department of Physical Education and Sports Teaching, Adiyaman University, 02000 Adiyaman, Turkey
5  Department of Coaching Education, Mersin University, 33000 Mersin, Turkey
*  Correspondence: cigdem_sahanli@hotmail.com; Tel.: +090-545-332-8401

**Abstract:** This study aimed to examine physical education teachers' perceptions of work alienation in Turkey according to different variables (including gender, marital status, school level, availability of a gym in the school, age, and years of service) during the COVID-19 pandemic, which affected sustainability in education on a global scale. The study employed the survey method and research data were collected from 442 volunteer physical education teachers working in different provinces of Turkey through the "Physical Education Teachers' Alienation to Work Scale". The results showed that physical education teachers had low levels of alienation in their work. The scale's subdimensions 'occupational isolation' and 'powerlessness' indicated higher levels of work alienation compared to other subdimensions. Among teachers who had completed their graduate education, the level of work alienation was higher in the subdimensions 'powerlessness' and 'occupational alienation'. Based on a comparison with prior research on sustainability in education, the COVID-19 pandemic could be said to have no significant impact on physical education teachers' levels of work alienation. The cause of work alienation among physical education teachers was structural issues rather than specific time-bound events such as the COVID-19 pandemic.

**Keywords:** work alienation; physical education teacher; COVID-19; sustainability in education

## 1. Introduction

Work alienation, a concept extensively studied in various fields with multiple definitions, has the potential to negatively influence both individual performance and organizational goals [1,2]. The experience of alienation is related to reduced employee participation and performance [3,4]. Work alienation refers to incompatibility between employee values and job roles in an organization and is associated with a disconnection from work both cognitively and emotionally [5–7]. This concept comprises various environmental and internal elements [5]. These factors not only lead to individual dissatisfaction but also result in the perception of work as meaningless, with serious individual and social consequences [8,9]. Work alienation has emerged as a product of modern society, affecting several domains of social life. The effects of alienation from work have been expressed in many areas ranging from working life to family life and from education to the media [5,8,10].

Education is one of the fields where the effects of work alienation may be more directly observed. In particular, the anti-democratic structure of the education system, intense curricula, physical inadequacy of schools and classrooms, the teaching of information that cannot be used in daily life, and similar factors can result in alienation in the field of education [11]. Just as an alienation from work emerged as a product of modern society,

alienation in education has also emerged as a product of modern education [12]. Teachers, as the key actors and employees in education, are among the groups most susceptible to the phenomenon of alienation in education [13]. Non-democratic administrative practices, inadequate school conditions, high student-teacher ratios, excessive curriculum demands, unappreciated labor, and meagre economic opportunities for teachers contribute to teachers' alienation from their work [11,12,14] (pp. 5–13).

In addition to such negative issues, physical education (PE) teachers also face additional challenges, including limited sports equipment and facilities, conflicting views of school administrators and parents, and the coordination of school sports teams [15,16].In addition, PE teachers play crucial roles in areas such as providing infrastructure for competitive sports, promoting health awareness, fostering students' understanding of health, and encouraging participation in sports among all individuals and promoting a lifelong engagement in physical activity. The weight of such wide-ranging responsibilities may lead to physical education teachers feeling more alienated from their school and work, as well as from the learning and teaching processes to a greater extent than other teachers.

The COVID-19 pandemic caused the temporary closure of educational institutions and the adoption of emergency distance education all around the world [17]. This situation imposed substantial new challenges on educational institutions, even in countries with adequate technological infrastructure. For example, teachers had to adapt to teaching remotely without sufficient training [18,19]. These conditions had a particularly significant impact on the delivery of hands-on school subjects such as physical education and sports [20] and placed extra pressure on PE teachers [21]. Therefore, the COVID-19 pandemic had potential effects that have stimulated scholarly discussion over the possibility of an increased alienation from work among PE teachers [22–24].

This study was conducted with a longitudinal approach to investigate whether the level of work alienation among physical education teachers in Turkey, as reported in 2013 [16], has changed as a result of the COVID-19 pandemic and to determine the current level of work alienation among physical education teachers during the pandemic. An additional aim of the study was to examine the variations in the subdimensions of work alienation for PE teachers such as feelings of meaninglessness, powerlessness, and alienation specific to PE teachers and occupational isolation, based on factors such as gender, marital status, school level, the availability of a gym in the school, age, and the number of years of service.

## 2. Materials and Methods

### 2.1. Research Design

The study employed the longitudinal research design, which is a quantitative research method [25] (p. 103). In this context, it aimed to examine whether the data obtained in a 2013 study conducted by Temel et al. [16] showed a different trend over the course of the COVID-19 pandemic [26] (p. 207).

### 2.2. Data Collection Tool

Research data were collected through the "Physical Education Teachers' Work Alienation Scale", developed by one of the authors of this study [16]. The scale was developed as part of the author's doctoral thesis in 2010 [27], and a face-to-face and online form of the scale was used to collect data to determine the level of work alienation among PE teachers. The findings concerning the analysis of those data were published in an article [16].

The scale consists of 38 items and four subdimensions. The subdimensions of the scale are as follows: *meaninglessness*, *powerlessness*, *alienation from physical education teaching*, and *occupational isolation*. The study investigated the impact of the COVID-19 pandemic on the work alienation levels among PE teachers, and the same measurement tool was used to collect data to address this research need.

*2.3. Participants*

The study sample consisted of 442 physical education (PE) teachers working in different provinces of Turkey during the 2020–2021 academic year, and the demographic characteristics of the teachers are presented in Table 1.

**Table 1.** The characteristics of participants.

| Gender | f | % |
|---|---|---|
| Male | 328 | 74.2 |
| Female | 114 | 25.8 |
| Total | 442 | 100.0 |
| **Age** | **f** | **%** |
| 21–30 | 98 | 22.2 |
| 31–40 | 140 | 31.7 |
| 41–50 | 184 | 41.6 |
| 51 and more | 20 | 4.5 |
| Total | 442 | 100.0 |
| **Years of service** | **f** | **%** |
| 1–10 | 190 | 42.99 |
| 11–20 | 143 | 32.35 |
| 21 and more | 109 | 24.66 |
| Total | 442 | 100.0 |
| **School** | **f** | **%** |
| Secondary | 259 | 58.6 |
| High school | 183 | 41.4 |
| Total | 442 | 100.0 |
| **Type of School** | **f** | **%** |
| Government | 400 | 90.5 |
| Private | 42 | 9.5 |
| Total | 442 | 100.0 |
| **Sport Facilities** | **f** | **%** |
| Yes | 153 | 34.6 |
| No | 289 | 65.4 |
| Total | 442 | 100.0 |

*2.4. Data Analysis*

To check whether the research data were normally distributed, the values of skewness and kurtosis were examined, and the data were found to be distributed between +3 and −3, thus it was decided that the data were normally distributed [28] (pp. 3–427) (Table 2). The study employed the horizontal scanning model. Data analyses were performed on the SPSS 26 software package. In the analysis, a t-test and one-way analysis of variance (one-way ANOVA) were used for pairwise comparisons (gender, marital status, school level, school type, and the availability of a gym) as well as for multiple comparisons (i.e., age and years of service). The Tukey test, one of the post hoc tests, was used to determine the difference between the groups revealed by the one-way ANOVA. The t-test can be used to determine if there is a significant difference between two groups in studies where a difference is predicted. However, to comment on the magnitude of this difference, the effect value must be known [29] (pp. 5–55). In cases where the group means varied significantly in the *t*-test, Cohen's d ($\delta$) formula was employed to calculate the effect size [29] (pp. 5–55). Cohen's d formula can be used to interpret the effect size of a study. A small effect is considered to be up to 0.2, a medium effect is up to 0.5, a high effect is up to 0.8, and a very large effect is up to 1.3 [30]. A p value less than 0.05 ($p < 0.05$) was considered statistically significant.

**Table 2.** Work alienation levels of participants.

|  | n | X | SS | Skewness | Kurtosis |
|---|---|---|---|---|---|
| Meaningless | 442 | 4.407 | 0.3680 | −0.556 | 0.136 |
| Powerlessness | 442 | 3.644 | 0.7040 | −0.340 | −0.167 |
| PE teacher alienation | 442 | 4.396 | 0.5973 | −1.375 | 1.724 |
| Occupational isolation | 442 | 3.552 | 0.7465 | −0.374 | −0.053 |

## 3. Results

The mean, standard deviation, kurtosis, and skewness values are given in Table 2. Upon an examination of the data, it was found that PE teachers had the highest levels of work alienation in the subdimensions of occupational isolation and powerlessness when compared to the other subdimensions.

According to the results in Table 3, the gender of PE teachers did not have a significant impact on the levels of work alienation in the subdimensions of meaninglessness, powerlessness, and alienation from a physical education lesson and occupational isolation.

**Table 3.** Comparison of Work Alienation Levels of Physical Education Teachers by Gender.

|  | Gender | N | X⁻ | SS | SD | T | P |
|---|---|---|---|---|---|---|---|
| Meaningless | Female | 114 | 4.357 | 0.3783 | 440 | 1.702 | 0.090 |
| | Male | 328 | 4.425 | 0.3633 | | | |
| Powerlessness | Female | 114 | 3.627 | 0.7058 | 440 | −0.310 | 0.757 |
| | Male | 328 | 3.650 | 0.7043 | | | |
| PE Teacher Alienation | Female | 114 | 4.360 | 0.5921 | 440 | −0.743 | 0.458 |
| | Male | 328 | 4.408 | 0.5995 | | | |
| Occupational isolation | Female | 114 | 3.628 | 0.7575 | 440 | 1.183 | 0.237 |
| | Male | 328 | 3.527 | 0.7422 | | | |

According to Table 4, the marital status of PE teachers did not have a significant impact on the levels of work alienation in the subdimensions of meaninglessness, powerlessness, occupational isolation, and alienation from a physical education lesson.

**Table 4.** Comparison of Work Alienation Levels of Physical Education Teachers by Marital Status.

|  | Marital Status | N | X⁻ | SS | SD | T | P |
|---|---|---|---|---|---|---|---|
| Meaningless | Single | 113 | 4.448 | 0.3278 | 442 | 1.38 | 0.168 |
| | Married | 329 | 4.393 | 0.3803 | | | |
| Powerlessness | Single | 113 | 3.713 | 0.6694 | 442 | 1.20 | 0.227 |
| | Married | 329 | 3.621 | 0.7149 | | | |
| PE teacher alienation | Single | 113 | 4.363 | 0.6000 | 442 | −0.67 | 0.502 |
| | Married | 329 | 4.407 | 0.5968 | | | |
| Occupational isolation | Single | 113 | 3.476 | 0.7015 | 442 | −1.24 | 0.212 |
| | Married | 329 | 3.578 | 0.7606 | | | |

Table 5 shows the level of the school where the PE teachers' work had no significant impact on the levels of work alienation in the subdimensions of meaninglessness, powerlessness, occupational isolation, and alienation from a physical education lesson.

**Table 5.** Comparison of Work Alienation Levels of Physical Education Teachers by School Level.

|  | School Level | N | X⁻ | SS | SD | T | P |
|---|---|---|---|---|---|---|---|
| Meaningless | Secondary | 259 | 4.403 | 0.3847 | 442 | −0.29 | 0.767 |
|  | High | 183 | 4.413 | 0.3439 |  |  |  |
| Powerlessness | Secondary | 259 | 3.676 | 0.7381 | 442 | 1.12 | 0.263 |
|  | High | 183 | 3.600 | 0.6520 |  |  |  |
| PE teacher alienation | Secondary | 259 | 4.409 | 0.5806 | 442 | −0.53 | 0.592 |
|  | High | 183 | 4.378 | 0.6212 |  |  |  |
| Occupational isolation | Secondary | 259 | 3.595 | 0.7438 | 442 | 1.43 | 0.151 |
|  | High | 183 | 3.491 | 0.7482 |  |  |  |

Table 6 shows a significant difference in the levels of work alienation between the PE teachers working in public schools and those working in private schools, particularly in the subdimensions of meaninglessness and powerlessness. Cohen's d was calculated to determine the effect size of the difference between the groups. Based on the method provided by Cohen in 1988 [29] (pp. 5–55), the statistical difference was found to have an effect size of 0.679 (Cohen's d value) in the meaninglessness subdimension and 0.474 in the weakness subdimension. When the other subdimensions were examined, no significant difference was found.

**Table 6.** Comparison of Work Alienation Levels of Physical Education Teachers by School Type.

|  | School Type | N | X⁻ | SS | SD | T | P | Cohen's d |
|---|---|---|---|---|---|---|---|---|
| Meaningless | Public | 400 | 4.387 | 0.3728 | 442 | −3.64 | 0.000 | 0.679 |
|  | Private | 42 | 4.602 | 0.2475 |  |  |  |  |
| Powerlessness | Public | 400 | 3.614 | 0.7035 | 442 | −2.84 | 0.005 | 0.474 |
|  | Private | 42 | 3.936 | 0.6459 |  |  |  |  |
| PE teacher alienation | Public | 400 | 4.400 | 0.5967 | 442 | 0.41 | 0.677 | 0.067 |
|  | Private | 42 | 4.359 | 0.6089 |  |  |  |  |
| Occupational isolation | Public | 400 | 3.561 | 0.7553 | 442 | 0.76 | 0.442 | 0.131 |
|  | Private | 42 | 3.468 | 0.6582 |  |  |  |  |

Table 7 shows that there was a significant difference in the work alienation levels in the powerlessness subdimension among PE teachers depending on whether or not their school had a gym. Cohen's d was calculated to determine the effect size of the difference between the groups in the independent groups t-test, which measures the magnitude of the difference between the groups in a statistical analysis. Based on the method provided by Cohen in 1988 [29] (pp. 5–55), the statistical difference was found to have an effect size of 0.316 in the weakness subdimension. When the other subdimensions were examined, it was determined that there was no significant difference.

Table 8 shows that there was a significant difference in the meaninglessness and occupational isolation subdimensions of work alienation among PE teachers based on their age. The Tukey test was used to determine which groups the difference was between and it was found that the difference was between group 1 (teachers aged 21–30) and group 2 (teachers aged 31–40) and it was also between group 1 and group 3 (teachers aged 41–50) in the meaninglessness subdimension, and between the groups 1 and 3 and groups 2 and 3 in the occupational isolation subdimension. No significant difference was found in the other subdimensions.

**Table 7.** Comparison of Work Alienation Levels of Physical Education Teachers According to Availability of a Gym in Their Schools.

|  | Availability of gym | N | $\bar{X}$ | SS | SD | T | P | Cohen's d |
|---|---|---|---|---|---|---|---|---|
| Meaningless | Yes | 153 | 4.412 | 0.3614 | 442 | 0.213 | 0.831 | 0.021 |
|  | No | 289 | 4.404 | 0.3721 |  |  |  |  |
| Powerlessness | Yes | 153 | 3.787 | 0.6675 | 442 | 3.13 | 0.002 | 0.316 |
|  | No | 289 | 3.569 | 0.7121 |  |  |  |  |
| PE teacher alienation | Yes | 153 | 4.368 | 0.6049 | 442 | −0.704 | 0.482 | 0.070 |
|  | No | 289 | 4.411 | 0.5937 |  |  |  |  |
| Occupational isolation | Yes | 153 | 3.571 | 0.7113 | 442 | 0.389 | 0.698 | 0.039 |
|  | No | 289 | 3.542 | 0.7655 |  |  |  |  |

**Table 8.** Comparison of Work Alienation Levels of Physical Education Teachers by Age.

|  | Age | N | $\bar{X}$ | SS | SD | ANOVA F | ANOVA P | Tukey |
|---|---|---|---|---|---|---|---|---|
| Meaningless | 21–30 | 98 | 4.519 | 0.309 | 442 | 4.375 | 0.005 | 1–2 |
|  | 31–40 | 140 | 4.386 | 0.389 |  |  |  |  |
|  | 41–50 | 184 | 4.359 | 0.37 |  |  |  | 1–3 |
|  | 51 and more | 20 | 4.450 | 0.331 |  |  |  |  |
| Powerlessness | 21–30 | 98 | 3.736 | 0.679 | 442 | 1.531 | 0.206 |  |
|  | 31–40 | 140 | 3.579 | 0.723 |  |  |  |  |
|  | 41–50 | 184 | 3.624 | 0.696 |  |  |  |  |
|  | 51 and more | 20 | 3.841 | 0.726 |  |  |  |  |
| PE teacher alienation | 21–30 | 98 | 4.475 | 0.574 | 442 | 1.733 | 0.160 |  |
|  | 31–40 | 140 | 4.316 | 0.643 |  |  |  |  |
|  | 41–50 | 184 | 4.401 | 0.573 |  |  |  |  |
|  | 51 and more | 20 | 4.527 | 0.545 |  |  |  |  |
| Occupational isolation | 21–30 | 98 | 3.407 | 0.740 | 442 | 5.323 | 0.001 | 1–3 |
|  | 31–40 | 140 | 3.438 | 0.781 |  |  |  |  |
|  | 41–50 | 184 | 3.687 | 0.704 |  |  |  | 2–3 |
|  | 51 and more | 20 | 3.816 | 0.656 |  |  |  |  |

Table 9 indicates that there was a significant difference in the meaninglessness subdimension of work alienation among physical education teachers based on their years of service. The Tukey test, a post hoc analysis, was used to determine which groups the difference was between, and it was found that the difference was between group 1 (teachers with 1 to 10 years of service) and group 2 (teachers with 11 to 20 years of service) in the meaninglessness subdimension. No significant difference was found in the other subdimensions.

**Table 9.** Comparison of Work Alienation Levels of Physical Education Teachers by Years of Service.

| | Years of Service | N | X⁻ | SS | SD | ANOVA | | Tukey |
| --- | --- | --- | --- | --- | --- | --- | --- | --- |
| | | | | | | F | P | |
| Meaningless | 1–10 | 190 | 4.465 | 0.366 | | | | |
| | 11–20 | 143 | 4.322 | 0.376 | 442 | 6.344 | 0.002 | 1–2 |
| | 21 and more | 109 | 4.418 | 0.340 | | | | |
| Powerlessness | 1–10 | 190 | 3.707 | 0.689 | | | | |
| | 11–20 | 143 | 3.575 | 0.678 | 442 | 1.496 | 0.225 | |
| | 21 and more | 109 | 3.625 | 0.757 | | | | |
| PE teacher alienation | 1–10 | 190 | 4.422 | 0.623 | | | | |
| | 11–20 | 143 | 4.335 | 0.556 | 442 | 1.100 | 0.334 | |
| | 21 and more | 109 | 4.430 | 0.602 | | | | |
| Occupational isolation | 1–10 | 190 | 3.473 | 0.777 | | | | |
| | 11–20 | 143 | 3.592 | 0.679 | 442 | 1.976 | 0.140 | |
| | 21 and more | 109 | 3.637 | 0.769 | | | | |

## 4. Discussion and Conclusions

Scholars have identified two key periods in the historical development of the concept of alienation. The first of these is the way that Hegel [31] and Marx [32] approached the concept, and the second is the modern exploration of alienation as it relates to human interactions and organizational work. This latter perspective has been studied extensively in empirical research, particularly by Fromm and American sociology and psychology circles following World War II. Fromm's perspective [33], which includes the concepts of powerlessness, meaninglessness, normlessness, isolation, and self-alienation, was later conceptualized by Seeman [34] and defined as the subdimensions of alienation. Finally, Temel et al. [16] defined the alienation of PE teachers from their work in four subdimensions: *meaninglessness*, *powerlessness*, *occupational isolation*, and *PE teacher's alienation*. The scale, its subdimensions, and the scale items were based on this background and literature.

This study explored the extent of work alienation among PE teachers during the COVID-19 pandemic and compared the results to those of a previous study conducted by Temel et al. in 2010 [16] prior to the pandemic. The focus of the study was on the subdimensions of work alienation among PE teachers, which include *meaninglessness*, *powerlessness*, *occupational isolation*, and *teachers' alienation from a physical education lesson*. As the education sector was heavily impacted by the pandemic, many educational activities had to be carried out via distance learning methods such as online technologies or television. When these distance education practices were evaluated, most PE teachers obviously suffered difficulties while conducting lessons, and so they tried to find new and creative solutions. Jeong and So [35] pointed out the difficulties of conducting online physical education classes in their study and stated that this method does not adequately reflect the value of physical education. They also emphasized the need to make changes in strategic learning methods and to specialize teachers in this subject in an attempt to overcome the difficulties brought about by the distance education methods. Cruickshank et al. [36] emphasize that teaching physical education online is challenging and, in many cases, physical education classes were not held. Instead, the format of physical activity lessons changed and teachers expressed concerns about student engagement and participation, preferring traditional in-person education.

Table 1 shows that PE teachers suffered higher levels of work alienation in the subdimensions of occupational isolation and powerlessness during the COVID-19 pandemic. This is consistent with the findings from previous research [16], indicating that physical education teachers generally had higher levels of work alienation in these subdimensions.



The cause of this is thought to be related to factors such as the management and administration of schools, the curriculum, and motivation. Specifically, physical education teachers feel powerless in areas such as having inadequate physical facilities in schools, a lack of sports equipment, and a lack of input in school administrative decisions.

In addition, PE teachers were also found to have higher levels of occupational isolation, which is one of the subdimensions of work alienation, as compared to other subdimensions. As a result, PE teachers who had higher levels of occupational isolation also suffered high levels of alienation from work [37–39]. Sulu et al. [1] stated that the lack of fair execution of work-related systemic processes and the failure of managers to engage in social relations with employees will cause occupational isolation and a sense of powerlessness. According to Siu [40], supporting the job satisfaction of employees and communicating effectively with them will not only eliminate the feelings of powerlessness but also contribute to job commitment.

In our study, we determined that the level of work alienation among the participants did not differ statistically according to the variables of gender, marital status, and the school they worked in. However, PE teachers' alienation from work was higher in the meaninglessness subdimension as compared to the variables of years of service during the COVID-19 pandemic (Table 9). As a PE teacher's years of service increase, the level of meaninglessness towards their profession also increases. When the data in Table 8 were examined, it was observed that the level of meaninglessness and occupational isolation increased with an increasing age. In their study examining the alienation levels among young teachers who are just starting their jobs and teachers who are further into their profession, Shoho and Nancy [41] (pp. 10–45) found that teachers with greater years of service had higher work alienation levels. This could be explained by the fact that young teachers who have just started to work may not have been affected by the negative conditions of both the education system and the current physical conditions of the school. PE teachers carry out the teaching process with expectations far from the reality of the system during their first years of work. However, in the following years, they cannot solve the systemic problems and become alienated from the work [42]. Tummers et al. [43] stated that employees' feeling that their work is meaningless is much more important than other subdimensions and emotions in terms of work efficiency and fulfilling the targeted behavior.

The findings of this study indicate that physical education teachers with a postgraduate education have higher levels of work alienation in the subdimensions of powerlessness, PE teacher's alienation, and occupational isolation. The traditional understanding of education, which Freire [44] (pp. 7–40) defined as the *banking method*, may be one of the reasons to be factored in. Today's schools distribute ideological knowledge and values, expand the market, control production, labor, and people, and help produce technical or business-oriented knowledge by creating widespread artificial needs within the population [45] (p. 64). Additionally, academic learning is the primary goal in today's educational institutions. All these factors tend to play a role in increasing teachers' alienation from work [12] (pp. 251–262).

According to the findings of the current study, when the individual is alienated from work, neither is the work done properly nor is the individual aware of this situation, as Marx puts it. For this reason, the behaviors and attitudes shown as a result of work alienation should not be taken into account because it is not the attitudes and behaviors of the individual that have the alienating effect here, but the work itself [46]. Therefore, it becomes evident that it is necessary to analyze the education system itself, not the physical education (PE) teachers.

When evaluated together with the research results [16] prior to the COVID-19 pandemic, it can be suggested that the pandemic did not have a significant effect on the work alienation levels of PE teachers; instead, their work alienation was primarily caused by inherent structural problems related to the current education system rather than the events spanning over a specific time period.

### 4.1. Limitations

The generalizability of the study findings to a wider population is limited by the educational practices adopted during the COVID-19 pandemic in Turkey, the time period of the research, and the data collected from the PE teachers who voluntarily participated in the research.

### 4.2. Implications

- PE teachers can receive on-the-job training for distance education methods.
- PE teachers can be empowered by ensuring their participation in the administrative decisions of the school.
- We also recommended that school administrations, PE teachers, and teachers of other subjects should work together to create more appropriate curricula.

**Author Contributions:** The author's contributions to this research are as follows: revealing the research problem C.T., M.K. and Ç.G.; collection of research data M.A.Y. and A.E.S.; methodology and data analysis S.U.; coordinating the process and responsible author Ç.G.; research findings and discussion S.U., T.T. and M.K.; spelling and stylistic arrangement M.A.Y., A.E.S. and C.T.; writing—review and editing, C.T., Ç.G., S.U., A.E.S., M.A.Y., M.K. and T.T. All authors have read and agreed to the published version of the manuscript.

**Funding:** This research received no external funding.

**Institutional Review Board Statement:** The study group of this research consisted of adult individuals. Ethics committee approval was not required, as a voluntary consent form was obtained from each participant.

**Informed Consent Statement:** Informed consent was obtained from all participants involved in the study.

**Data Availability Statement:** Data are not available because of privacy of participants.

**Acknowledgments:** The authors thank the PE teachers working for the Ministry of National Education.

**Conflicts of Interest:** The authors declare no conflict of interest.

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
