# Peer review of "The Impact of COVID-19 Process on Sustainability in Education: Work Alienation of Physical Education and Sports Teachers"

_sustainability, doi:10.3390/su15032047_

Round 1

Reviewer 1 Report

Although the article is coherent and contains all important parts of the study design, I would suggest including the following points:

- More succinct explanation why the study is needed (is it useful to replicate studies? - include this point),

- More succinct explanation why readers in other countries should know about the study. The author(s) mention physical conditions related to school which doesn't have to be the case in some of the countries,

- Weaknesses (limitations) of the study are missing. 

- How about indicating a few implications for reducing the alienating effect for physical education teachers?

Minor comments: 

- Please, avoid starting sentences with 'because'(p. 2, p. 11),

- Wrong use of semi-colons (Abstract). 

Author Response

Dear Reviewer, first of all, thank you for your review. The changes made to your comment are as follows;

-More succinct explanation why the study is needed (is it useful to replicate studies? - include this point); explanations for this suggestion 69-77. located between the lines.

-More succinct explanation why readers in other countries should know about the study. The author(s) mention physical conditions related to school which doesn't have to be the case in some of the countries; explanations for this suggestion 268-274. located between the lines.

-Weaknesses (limitations) of the study are missing; explanations for this suggestion 280-283. located between the lines.

-How about indicating a few implications for reducing the alienating effect for physical education teachers?; explanations for this suggestion 284-289. located between the lines.

-Please, avoid starting sentences with 'because'(p. 2, p. 11); explanations for this suggestion 234. located line, the "because" used on page 2 has been removed.

-Wrong use of semi-colons (Abstract). Fixed the use of semi-colons.

Reviewer 2 Report

The article entitled The Impact of Covid-19 Process on Sustainability in Education: Work Alienation of Physical Education and Sports Teachers is interesting first of all because it addresses a topic of high present interest: the effects of COVID-19 pandemic on the education system, which during the pandemics was challenged to adapt to the online environment with more or less success. The authors underline the fact that this process was not easy, especially when it comes to physical education.

The article contains the appropriate structure. It is correctly divided into relevant sections and their content coincides with their titles. Bibliography is correctly formulated.

My main recommendations are the following:

Title: It is too general. In fact, the article focusses on a case study (Turkey). I suggest including this in the title.

Abstract – The authors should clarify the objectives of the study. I see the results presented extensively, but not the objectives / aims of the paper.

Introduction – Has the topic been addressed before? If yes, please expand the literature review. Also: lines 69 – 70: “This research was carried out to determine the level of work alienation of physical education teachers in 2010” – I think this statement needs adjusting. I do not think that the authors carried out this research to determine the level of work alienation in 2010…but in 2020-2021.

Materials and Methods - Please clarify the methodological framework by including information on the programs used to process the data. Also…do the models used have any limitations?

Discussion and conclusions – It would be interesting to find out if there are differences in the results if we analyze the data not only by gender / age / type of school but also by residential environment (rural / urban) and geographical distribution (poorer / richer regions). The conclusions are very vague, they should get more consistency, underline the added value of the paper.

Author Response

Dear Reviewer, first of all, thank you for your review. The arrangements and explanations made in line with your suggestions are as follows;

-"Title: Very general.  In fact, the article focuses on maintaining a case (Turkey).  We recommend reading it in the title."; The research was conducted in Turkey and was collected from its connections in Turkey. However, alienation from the profession is considered a universal organ and in terms of alienation from the professions of those who live in different countries from the organs obtained.

-"Abstract---Authors should clarify the aims of the study. I see the results presented comprehensively, but I do not see the aims/objectives of the article."; explanations for this suggestion 18-21. located between the lines. 

-"Introduction – Has the topic been covered before? If yes, please expand the literature review. Also: Lines 69 – 70: “This research was conducted in 2010 to determine the level of alienation of physical education teachers from work” – I think this statement needs to be corrected. I think that the authors conducted this research in 2020-2021, not in 2010, to determine the level of work alienation."; This study was first started in 2010 with the development of a measurement tool to determine the level of alienation of physical education teachers from their profession. Then, with the measurement tool developed in 2010, data were collected in 2013 from physical education teachers and it was determined that the teachers were alienated from their profession. With the same measurement tool, data were collected in the 2020-2021 academic year during the Covid-19 period and similar results were found. As a result, it has been evaluated that even an epidemic period such as Covid-19 that affects all education does not have different effects on the alienation of physical education teachers from their profession and that alienation is caused by problems related to the education system. In addition, the necessary regulation is located on line 79.

-"Materials and Methods   Please clarify the methodological framework by adding information about the programs used to process the data. Also… are there any limitations to the models used?"; Data analysis and Data collection tools sections have been rearranged in line with your suggestions. Analysis programs and related explanations have been added to the relevant sections. There is also an additional section on limitations at 280-283. lines located in the study. 

-"Discussion and results---If we analyze the data not only by gender/age/type of school, but also by residential setting (rural/urban) and geographic distribution (poorer/richer regions), it will be interesting to find out if there are differences in results). Results are too vague, should be more consistent, underline the added value of the article."; Your suggestion is very valuable to us. However, in Turkey, all teachers are responsible for teaching the same curricula and practices under the Ministry of National Education. Although the responsibilities of teachers are the same, schools in our country are divided into private and public schools. Therefore, we took these distinctions into consideration when determining the variables. However, in the future, considering your suggestion, we may plan to conduct a study in which the variables of "settlement environment (rural/urban) and geographical distribution (poorer / richer)" will be used. We are already wondering what the results will be.

Reviewer 3 Report

Thankyou for this submission. Some revision work is required at this stage. Please can you address the following points:

At the outset, something explaining the context and aims of the study, as well as the choice of physical education teachers as the sample would be helpful. Lines 60-68 give some background to the issues for these teachers, but not why this study includes them in particular.

In addition, more attention is needed regarding the significance of covid-19, both in the introduction and throughout the paper. This might be addressed in part by placing the detail in the paragraph from lines 168-181 earlier in the piece. However there must be more intentional correlation through the article between the research aims including exploring the impact of the pandemic, and the research.

The research aims need to be presented more fully and explicitly.

Please provide a Literature Review to locate this study in a wider academic context. 

What is the rationale behind the particular items and sub-dimensions? How do they particularly evidence concern with the research questions?

Ethical considerations should be addressed. 

Line 47 - change 'work life was seen' to 'might be evidenced.' 

How was data acquired? A more explicit outline of the methodology, with acknowledgement of limitations, is required here. 

Please explain 't-test for pairwise comparisons and one-way analysis 100 of variance (ANOVA) for multiple comparisons were used.' More detail needed here for readers unaware of this method - lines 100-1

Please explain 'Cohen's d and Tukey methods' - line 101

The discussion is interesting but would benefit from again more detail, referenced and with recommendations for future practice and research. The detail about Marx is interested and might be expanded somewhat, with more discussion on the philosophical dimension of this subject, especially in relation to educational philosophy and/or critical pedagogy.  

Address issues with punctuation and some English grammar need attention - possibly before the proof-reading stage. 

I look forward to reading the revised article. 

Author Response

Dear Reviewer, first of all, thank you for your review. The changes made to your comment are as follows;

-"At the beginning, something that explains the context and objectives of the study, as well as the selection of physical education teachers as samples, would be helpful. Lines 60-68 give some background to the problems for these teachers, but this is not why the study specifically includes them"; explanations for this suggestion 69-77. located between the lines.

-"In addition, more attention is needed regarding the significance of covid-19, both in the introduction and throughout the paper. This might be addressed in part by placing the detail in the paragraph from lines 168-181 earlier in the piece. However, there must be more intentional correlation through the article between the research aims including exploring the impact of the pandemic, and the research."; explanations for this suggestion 190-209. located between the lines.

-"The research aims need to be presented more fully and explicitly."; explanations for this suggestion 78-85. located between the lines.

-"Please provide a Literature Review to locate this study in a wider academic context. "; explanations for this suggestion 69-77. located between the lines. In addition, the discussion section has been expanded in this direction.

-"What is the rationale behind the particular items and sub-dimensions? How do they particularly evidence concern with the research questions?";

Scholars focus on two periods related to the historical development of the concept of alienation. The first of these is the way Hegel and Marx handle alienation, and the other is evaluated within the framework of modern life-human interaction and organizational work, which has been examined by experimental (empirical) studies, especially by Fromm and American sociology and psychology circles after World War II. From this point of view, the concepts of powerlessness, meaninglessness, normlessness, isolation and self-alienation, conceptualized by Seeman (1959) and defined as the dimensions of alienation, were discussed and finally Temel (2013) defined physical education teachers' alienation in their work in four sub-dimensions -meaninglessness, powerlessness, vocational isolation and physical education teacher’s aliention. The scale, its sub-dimensions and scale items are based on this background and literature.

-"Ethical issues must be addressed."; We have received the ethics committee approval dated 01.03.2021 and numbered 2021PO75 by the Ethics Committee of Akdeniz University Institute of Social Sciences. The editor has been informed about this issue.

-"Line 47 - change 'work-life seen' to 'provable'."; explanations for this suggestion 50-51. located between the lines.

-“How was the data obtained? A clearer outline of the methodology and acceptance of limitations is required here.”; explanations for this suggestion 280-283., 109-125. and 93-98. located between the lines.

-“Please explain 't-test for pairwise comparisons and one-way analysis 100 of variance (ANOVA) for multiple comparisons were used.' More detail needed here for readers unaware of this method - lines 100-1"; explanations for this suggestion 114-120. located between the lines.

-"Please explain 'Cohen's d and Tukey methods' - line 101"; explanations for this suggestion 120-125. located between the lines.

-"The discussion is interesting but would benefit from again more detail, referenced and with recommendations for future practice and research. The detail about Marx is interested and might be expanded somewhat, with more discussion on the philosophical dimension of this subject, especially in relation to educational philosophy and/or critical pedagogy.  "; explanations for this suggestion 259-268. located between the lines.

-"Address issues with punctuation and some English grammar need attention - possibly before the proofreading stage."; Necessary controls have been made on the entire text.

Round 2

Reviewer 2 Report

The article can be accepted in present form.

Author Response

Dear Reviewer,

Thank you for your reviews.

Reviewer 3 Report

Thankyou for addressing the points raised in my first review.

Please can you include the following (taken from your response to myself) in the next iteration of the article as I feel that this important detail is necessary before publication. 

Author Response

Dear Reviewer, first of all, thank you for your review. We have included the explanation we have made for you in the article. You can see the changes made in the attached text between lines 190-201.
